# ON THE ROLE OF FORGETTING IN FINE-TUNING REINFORCEMENT LEARNING MODELS

**Maciej Wołczyk**
Jagiellonian University

**Bartłomiej Cupiał**
Jagiellonian Unviersity
Ideas NCBR

**Michał Zając**
Jagiellonian University

**Razvan Pascanu**
DeepMind

**Łukasz Kuciński**
Ideas NCBR,
Polish Academy of Sciences

**Piotr Miłoś**
Ideas NCBR,
Polish Academy of Sciences,
deepsense.ai

## ABSTRACT

Recently, foundation models have achieved remarkable results in fields such as computer vision and language processing. Although there has been a significant push to introduce similar approaches in reinforcement learning, these have not yet succeeded on a comparable scale. In this paper, we take a step towards understanding and closing this gap by highlighting one of the problems specific to foundation RL models, namely the data shift occurring during fine-tuning. We show that fine-tuning on compositional tasks, where parts of the environment might only be available after a long training period, is inherently prone to catastrophic forgetting. In such a scenario, a pre-trained model might forget useful knowledge before even seeing parts of the state space it can solve. We provide examples of both a grid world and realistic robotic scenarios where catastrophic forgetting occurs. Finally, we show how this problem can be mitigated by using tools from continual learning. We discuss the potential impact of this finding and propose further research directions.

## 1 INTRODUCTION

Foundation models are one of the most important trends in deep learning in recent years. Pre-training a massive model on a very wide array of diverse data and then fine-tuning it on a specific task proved to be a highly useful paradigm. Models such as BERT (Devlin et al., 2019), GPT-3 (Brown et al., 2020), and CLIP (Radford et al., 2021) significantly outperformed their predecessors in their respective benchmarks and enabled efficient adaptation to specific downstream tasks through the relatively cheap fine-tuning procedure. Even straightforward fine-tuning approaches work very well in this paradigm, as it is enough to simply use the pre-trained model as the initialization and train for a short amount of time. Although this approach achieved remarkable results in fields like natural language processing, computer vision, automatic speech recognition (Radford et al., 2022) and cheminformatics (Chithrananda et al., 2020), it has not yet succeeded on a similar scale in reinforcement learning.

There has been a significant push towards foundation models in reinforcement learning, with different approaches focusing on building transferable models of the world (Seo et al., 2022; Sun et al., 2022), learning reusable features (Schwarzer et al., 2021; Stooke et al., 2021) or obtaining general policies through offline RL (Kumar et al., 2022). Although these efforts lead to significant advances in this field and remarkable achievements (Brohan et al., 2022; Adaptive Agent Team et al., 2023), the problem of building foundation models in reinforcement learning seems to be inherently more difficult than in supervised learning. In this work, we attempt to highlight and analyze one of the reasons for this difficulty gap, namely the data shift that naturally occurs during fine-tuning RL models.

In most supervised learning scenarios, the fine-tuning data is completely static and does not change during the training. On the other hand, even if the environment used for the purpose of fine-tuning is

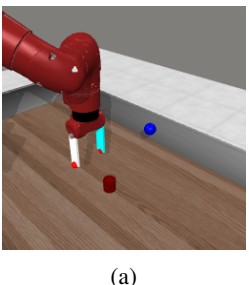 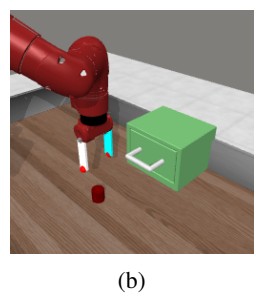 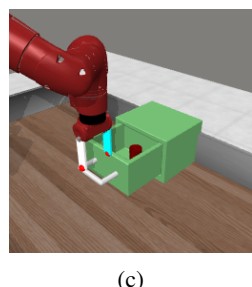

(a)                                        (b)                                        (c)

Figure 1: Example of the forgetting problem. (a) We start with a pre-trained model that is capable of picking up and placing objects. (b) In the downstream task, we need to pick up the object inside the drawer which is currently closed. The model needs to learn how to open the drawer first. (c) If we simply fine-tune our pre-trained model on this task, it will forget how to pick up and place objects. After opening the drawer, it will need to re-learn how to move objects, therefore limiting the usefulness of the pre-trained model.

completely stationary, the data the model sees while training might not be. If some parts of the state space are not available in the initial phase of fine-tuning (e.g. due to exploration policy not being able to reach there), the model will focus on learning only the parts it sees. Even if the pre-trained model implements an optimal policy on this unseen part of the environment, its parameters will be effectively overwritten by the current task, in a phenomenon called catastrophic forgetting (French, 1999). By the time we are able to visit the previously unseen parts of the environment, the pre-trained knowledge will be completely lost. As such, the knowledge transfer might be limited only to the parts of the environment which are available from the start. An example of such a situation is shown in Figure 1.

Similar considerations appear in existing papers on fine-tuning reinforcement learning models. In particular, Baker et al. (2022) mention that they include a regularization term to limit forgetting. Other strategies commonly applied in fine-tuning RL which might implicitly help with catastrophic forgetting include mixing new data with the old data (Kumar et al., 2022) and introducing modularity to the model (Seo et al., 2022). As such, we do not claim to be the first ones who noticed this issue. Instead, we focus on the characterization and the experimental analysis of the problem of forgetting in fine-tuning RL models, as well as connecting this problem to the vast continual learning literature. We show that existing continual learning approaches can largely alleviate forgetting.

This paper is a preliminary study of this phenomenon, to be updated and extended in the future. Our contributions are as follows:

- Describing and formalizing the forgetting problem in fine-tuning reinforcement learning models, along with a synthetic example.

- Analyzing the problem through an experimental study of a compositional robotic task.

- Applying continual learning methods on this problem to show they help with alleviating the forgetting issue.

## 2 PRELIMINARIES

### 2.1 REINFORCEMENT LEARNING

We follow the standard Markov Decision Process (MDP) formulation for reinforcement learning. We define MDP as a tuple $\mathcal{M} = (\mathcal{S}, \mathcal{A}, p, R, \gamma)$, where $\mathcal{S}$ is the state space, $\mathcal{A}$ is the action space, $p : \mathcal{S} \times \mathcal{A} \to P(\mathcal{S})$ are the state transition probabilities, $R : \mathcal{S} \times \mathcal{A} \to \mathbb{R}$ is the reward function, and $\gamma \in [0, 1]$ is the discount factor. The agent interacts with the environment through the policy $\pi : S \to P(\mathcal{A})$ with the goal of maximizing the expected reward defined as $\mathbb{E}[\sum_{t=0}^{\infty} \gamma^t r_t]$, where $r_t$ is the reward at timestep $t$. Additionally, we define the occupancy measure $\mu_\pi(s) = \sum_{t=0}^{\infty} \gamma^t P(s_t = s)$ to capture the visitation probabilities of each state under policy $\pi$.

## 2.2 Continual Learning

The field of continual learning focuses on the problem of learning from a changing stream of data and, in particular, dealing with catastrophic forgetting. As such, we want to investigate the usefulness of continual learning tools in our setting. Three major families of continual learning methods include regularization-based approaches, which limit changes in the parameters of the network, replay-based methods, which rehearse data from the past to minimize forgetting, and modularity-based methods which try to keep separate parts of the network for each encountered task (De Lange et al., 2021). Here, we will focus on the first two families, and we leave the investigation of modularity for future work. We choose established and well-tested methods for our study. Although classical continual learning approaches primarily focus on constantly minimizing forgetting throughout the training, here we will focus only on protecting the knowledge of the pre-trained model. This requires only small adjustments to existing methods.

**Regularization-based** In this family, we consider L2 and Elastic Weight Consolidation (EWC) (Kirkpatrick et al., 2017), which apply a quadratic penalty on parameter changes in order to mitigate forgetting. In our case, this is equivalent to introducing an auxiliary loss of the form: $R(\theta) = \sum_i F^i(\theta^i_{\text{pre}} - \theta)^2$, where $\theta_{\text{pre}}$ are the weights of a pre-trained model, $F^i$ are weighting coefficients, and the sum iterates over all parameters in the model. For L2, $F = 1$, as we simply minimize the Euclidean distance between the old and new parameters. For EWC, $F$ is the diagonal of the Fisher matrix, which approximates how small changes in each parameter impact the output distribution.

**Replay-based** We use behavioral cloning (BC) as a simple replay-based approach that worked well in previous works (Rebuffi et al., 2017; Wolczyk et al., 2022). In our case, behavioral cloning relies on distilling the knowledge from the pre-trained model into the model being fine-tuned by minimizing the Kullback-Leibler divergence between these two. However, since we initialize the fine-tuning model using the parameters of the pre-trained network, the KL loss effectively mitigates the loss of performance rather than introducing new knowledge. It is important to note that distillation-based approaches have been often used for transferring knowledge in RL (Rusu et al., 2015; Schmitt et al., 2018). We hypothesize that their success might be also understood from the perspective of avoiding forgetting.

In practice, we implement BC in the following way. Before the training, we gather a buffer of data $\mathcal{B}$ from the parts of the environment the model was pre-trained on. For each state-action tuple in this buffer, we save the action distribution $\pi^*(s)$ returned by the actor from the pre-trained model. During the training, we apply an auxiliary loss of the form $R(\theta) = \mathbb{E}_{s \sim \mathcal{B}}[D_{KL}(\pi(s) \parallel \pi^*(s))]$. In principle, we could also mitigate forgetting in the critic, but we find empirically that it hurts the performance, which is consistent with prior work (Wolczyk et al., 2022).

## 3 Background and problem outline

In this section, we define the data-shift problem in fine-tuning reinforcement learning models that leads to forgetting. The key distinction between fine-tuning in supervised learning and reinforcement learning is that the data for supervised fine-tuning is usually stationary. The same is not true for reinforcement learning. Even if the training environment itself is stationary, the policy we use for exploration constantly evolves, and, consequently, the distribution of the states we visit is changed. In particular, if a pre-trained model is capable of solving part of the environment which will only be encountered after prolonged training, its performance on that part might drop significantly during the training.

## 3.1 Compositional MDPs

Below, we formally describe how forgetting might occur in certain MDPs which we call compositional MDPs. Our goal is to solve MDP $\mathcal{M} = (\mathcal{S}, \mathcal{A}, p, R, \gamma)$, starting from a policy $\pi$ pre-trained on MDP $\tilde{\mathcal{M}} = (\tilde{\mathcal{S}}, \mathcal{A}, \tilde{p}, \tilde{R}, \gamma)$. We assume that $\tilde{\mathcal{M}}$ is a subset of the target MDP $\mathcal{M}$, that is, $\tilde{\mathcal{S}} \subset \mathcal{S}$, $\tilde{R} = R|_{\tilde{\mathcal{S}} \times \mathcal{A}}$, $\tilde{p}(\tilde{s}, a)(\tilde{t}) = \frac{p(\tilde{s}, a)(\tilde{t})}{\sum_{\tilde{u} \in \tilde{\mathcal{S}}} p(\tilde{s}, a)(\tilde{u})}$ for $\tilde{s}, \tilde{t} \in \tilde{\mathcal{S}}, a \in \mathcal{A}$, and that $\tilde{\mathcal{S}}$ contains high-value states

that we need to visit in order to solve the task. Finally, we assume that getting to $\tilde{\mathcal{S}}$ is not trivial, i.e. $\mu_\pi(\tilde{s}) \approx 0$ for any $\tilde{s} \in \tilde{\mathcal{S}}$ and a policy $\pi$ below some performance threshold $T \gg V_R$, where $V_R$ is the average performance of the random policy.

Assume that we have a pre-trained parametric policy $\pi_\theta$ that is nearly optimal on $\tilde{\mathcal{M}}$, but was not trained on $\mathcal{S} \setminus \tilde{\mathcal{S}}$. As such, it cannot be used to reach states $\tilde{\mathcal{S}}$ reliably, that is $\mu_{\pi_\theta}(\tilde{s}) \approx 0$, for all $\tilde{s} \in \tilde{\mathcal{S}}$. We can fine-tune the policy $\pi_\theta$ on the target task. However, since the problem requires training the model on a distribution that was not previously encountered, there is a significant risk of catastrophic forgetting. Namely, at the point when $\pi_\theta$ manages to reach the states it was pre-trained on, i.e. $\mu_{\pi_\theta}(\tilde{s}) \gg 0$ for some $\tilde{s} \in \tilde{\mathcal{S}}$, it might no longer be close to the optimal policy on $\tilde{\mathcal{M}}$.

The above formulation fits various environments that follow the temporal compositionality principle, i.e. they require a sequence of tasks to be executed. For example, the sub-MDP $\tilde{\mathcal{M}}$ might represent later levels of Montezuma's Revenge, advanced phases of the Minecraft challenge (i.e. using a pickaxe to dig diamonds), or certain skills in a robotic environment (e.g. picking and placing objects in Figure 1). In all of these settings, using a policy $\pi_\theta$ which is optimal on $\tilde{\mathcal{M}}$ might lead to interference with fine-tuning on $\mathcal{M}$, which in turn leads to losing good performance on $\tilde{\mathcal{M}}$.

### 3.2 TOY EXAMPLE

In order to illustrate the problem described above, we introduce the APPLERETRIEVAL$(c, M, T)$ environment, where $c, M, T$ denote important design parameters. We show that even simple RL algorithms with linear function approximators exhibit forgetting in this environment.

In APPLERETRIEVAL, visualized in Figure 2 the agent lives on a 1D gridworld. Starting at home at position $x = 0$, in Phase 1, the agent has to go to $x = M, M \in \mathbb{R}^+$ to retrieve the apple. Then, in Phase 2, the agent has to go back home by returning to $x = 0$. If not solved, the episode finishes after $T$ steps. The state is a vector $s = [1, -c]$ in Phase 1 and $s = [1, c]$ in Phase 2. The first element of the vector is just a constant[1] and the second encodes the information about the current Phase. As such, it is trivial to construct the optimal policy: go right during the first phase and go left during the second phase.

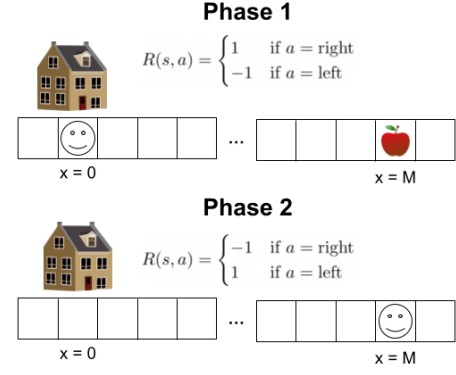

Figure 2: A schematic of the APPLERE-TRIEVAL enviornment.

We try to solve this environment using a linear model $\pi_w(s) = w^T s$ trained with the REINFORCE algorithm (Williams, 1992). We initialize its weights with a model that was only trained on Phase 2. For simplicity, we set $T = 100$ in all experiments. We show experimentally, that for certain values of the design parameters, APPLERETRIEVAL is an example of a compositional MDP, with $\tilde{\mathcal{M}}$ representing Phase 2 and $\mathcal{M}$ representing the whole environment. In particular, Figure 3a shows that for high enough distance $M$ the probability of getting to Phase 2 with a poor policy is very low. This, in turn, leads to high forgetting and overall poor performance.

Additionally, in this simple linear case, we can pinpoint the cause of the interference by looking at the weights of the model. If the pre-trained policy mostly relies on $s_2$ which represents the phase (i.e. $|w_2| \gg |w_1|$) then the interference will be limited. However, if the model relies on $s_1$ (i.e. $|w_2| \ll |w_1|$) then interference will occur as its value is the same in both phases. We can guide the model towards focusing on one or the other by changing the relative magnitude of $\frac{s_1}{s_2}$ by setting the $c$ parameter. The results presented in Figure 3b confirm our hypothesis, as lower values of $c$ encourage models to rely more on $s_1$ which leads to forgetting. Such low-level analysis is infeasible for deep

---

[1]Since we use a linear model without bias, it effectively serves as bias. We can omit it if we assume an affine model.

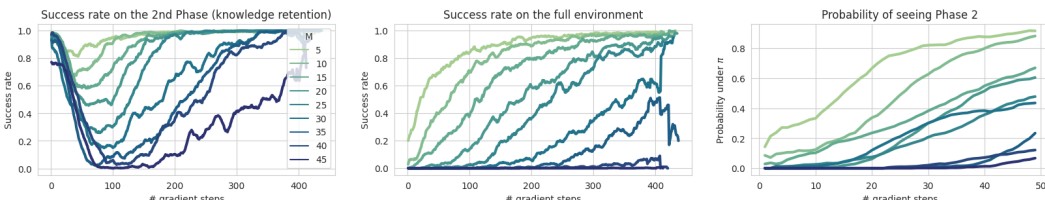

(a) Impact of $M$ on the results for set $c = 0.5$. Forgetting (left) becomes much more problematic as we increase the distance from the house to the apple, which in turn impacts the success rate on the whole environment (center). This happens since the probability of seeing Phase 2 in early training decreases (right, note x-scale change), leading to a stronger data shift.

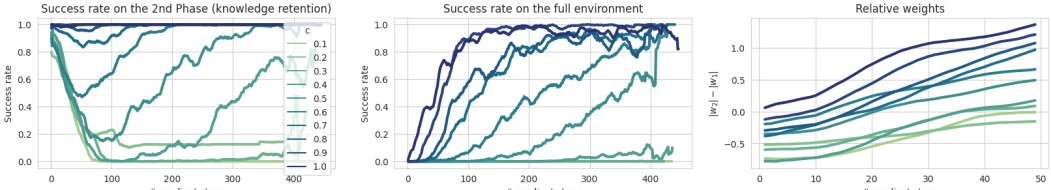

(b) Impact of $c$ on the results for set $M = 30$. For smaller $c$ forgetting (left) is greater and the overall success rate is smaller (center), since it encourages the pre-trained model to pay more attention to the task-shared variable $s_1$, as confirmed by looking at weight difference early in fine-tuning (right).

Figure 3: Study of the impact of the design parameters on the APPLERETRIEVAL results.

neural networks, but experimental results confirm that interference occurs in practice (Kirkpatrick et al., 2017; Kemker et al., 2018; Ramasesh et al., 2022).

Although ultimately we are interested in investigating types of phenomena on a much larger scale, this simple toy environment shows that the problem of forgetting can be in fact fundamental to fine-tuning. An anonymized version of an interactive version of this environment is available at `https://huggingface.co/spaces/LLParallax/Apple`. It allows users to train on different variants of APPLERETRIEVAL and observe forgetting. We hope that this tool will help with building basic intuitions for the issue of forgetting in RL fine-tuning.

## 4 EXPERIMENTS IN A ROBOTICS ENVIRONMENT

In this section, we experimentally investigate the catastrophic forgetting phenomenon in fine-tuning reinforcement learning models. We move on from the toy environment to a substantially more complex compositional robotic manipulation task and we use multi-layer neural networks as function approximators for the policy and the Q-value function. For all experiments, we use the Soft Actor-Critic (SAC) algorithm (Haarnoja et al., 2018a) and we defer technical details to Appendix A. All experiments in this section are run with at least 10 seeds.

### 4.1 STITCHEDENV

We use the Continual World benchmark (Wołczyk et al., 2021) which relies on Meta-World (Yu et al., 2020) environments as the testbed for our experiments. Continual World was recently proposed for investigating continual reinforcement learning. It introduces pre-defined sequences of environments where the tasks switch at set intervals. Although consistent with classical continual learning formulations, this approach does not properly cover the fine-tuning issue where the shift in the data is guided by the changes in the policy.

---

**Algorithm 1:** STITCHEDENV

**Input:** list of $N$ environments $E_k$, policy $\pi$, time limit $T$.
**Returns:** number of solved environments.
$i = 1; t = 1$ {Initialize env idx, timestep counter}
**while** $i \leq N$ **and** $t \leq T$ **do**
    Take a step in $E_i$ using $\pi$
    **if** $E_i$ is solved **then**
        $i = i + 1; t = 1$ {Move to the next env, reset timestep counter }
    **end if**
**end while**
**return** $i - 1$

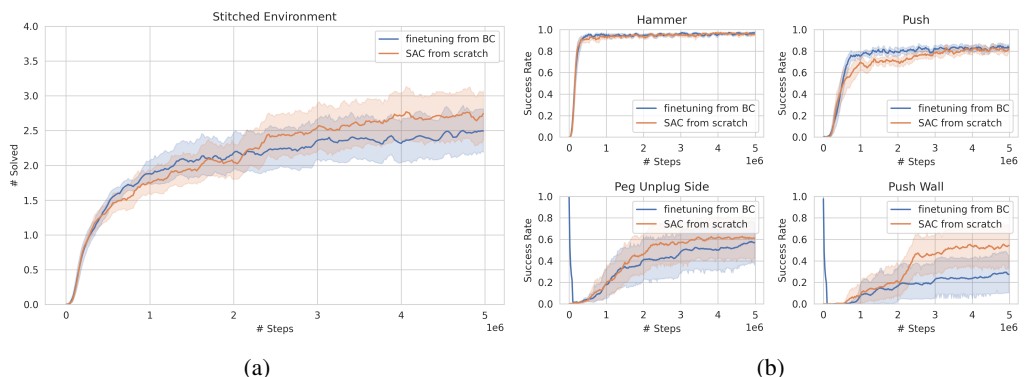

(a)  (b)

Figure 4: The performance of the fine-tuned model on STITCHEDENV compared to the performance of a model trained from scratch. (a) The average success of the agent throughout the training, i.e. how many tasks it can solve on average in a single episode. (b) The success rate of each of the tasks evaluated separately. The fine-tuned model rapidly forgets how to solve PEG-UNPLUG-SIDE and PUSH-WALL, the tasks it was pre-trained on, and then takes a long time to re-learn them. As such, using the pre-trained model does not lead to any improvements over the baseline here.

Instead, we use the tasks defined in Continual World to create a new environment dubbed STITCHEDENV. In STITCHEDENV each episode consists of a sequence of tasks stitched together, where the new task starts once the previous one is completed. Since each Continual World task has a clearly defined success condition (e.g. the object was placed properly), we use it as a signal when to move on to the next task. As such, in opposition to classical continual learning formulations, the data shift occurs here both on the level of a single trajectory and between trajectories. The proposed environment, therefore, fulfills the compositionality condition mentioned in the previous section. In this case, $\mathcal{M}$ would be the full environment, and $\tilde{\mathcal{M}}$ the later tasks in the sequence. The algorithm formally describing the behavior of STITCHEDENV is shown in Algorithm 1.

In our experiments, we will focus on a STITCHEDENV consisting of the following Continual World tasks: [`hammer`, `push`, `peg-unplug-side`, `push-wall`]. We use a model pre-trained on the last two tasks. The first two tasks were chosen to be significantly easier than the second two so that having a pre-trained policy on those would be very helpful. To assure that a multi-task solution exists, we present the model with the task ID of the current task. This allows us to simplify the experimental setting. We leave the task-agnostic formulation for future work.

## 4.2  FORGETTING

To investigate the forgetting problem, we use a model pre-trained on the last two tasks from the sequence, PEG-UNPLUG-SIDE and PUSH-WALL. For this purpose, we simply train a SAC agent in the multi-task regime[2] on these tasks until it can solve these environments perfectly (100% success rate). As such, if we were to run the obtained policy on STITCHEDENV(PEG-UNPLUG-SIDE, PUSH-WALL), it would also achieve perfect results. However, the full STITCHEDENV(HAMMER, PUSH, PEG-UNPLUG-SIDE, PUSH-WALL) proves much more difficult. The pre-trained policy, not having seen HAMMER or PUSH previously, is not able to solve these without training, and as such does not reach the state spaces from PEG-UNPLUG-SIDE and PUSH-WALL which come later in the sequence.

We train SAC on this environment for 5M steps, using the parameters of the actor and the critic from the pre-trained models as initialization. Figure 4a shows the performance of the algorithm throughout the training, compared to a baseline model which learns completely from scratch. Surprisingly, the pre-trained model does not perform better than the randomly initialized one, even though it con-

---

[2]We argue that the choice of pre-training is not crucial here and we might as well use an offline RL pre-training technique.

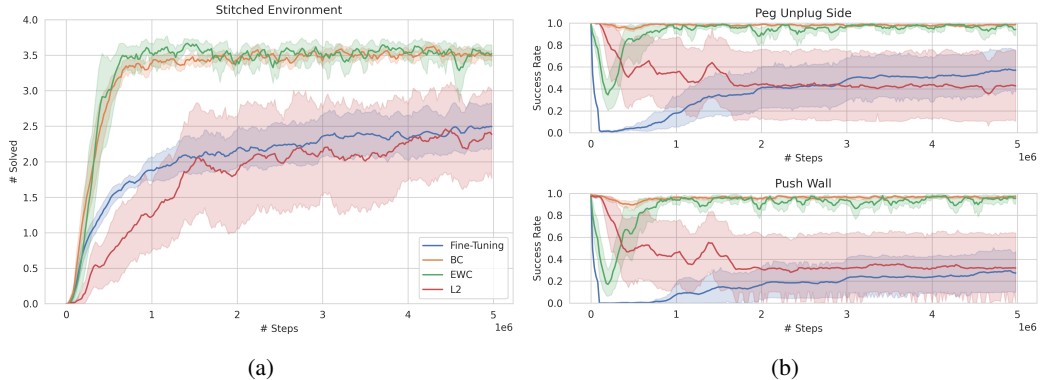

Figure 5: Results on STITCHEDENV for different continual learning methods. (a) By preserving the performance on the pre-trained tasks, we are now able to benefit from fine-tuning and achieve near-perfect results with EWC and behavioral cloning. (b) Performance on the tasks the model was pre-trained on. Continual learning methods are able to maintain or very quickly regain performance on these tasks.

tains information on how to solve two out of four tasks. We check the performance of the model on each task in separation by running the policy in single-task scenarios. Figure 4b shows that although the performance of the model on PEG-UNPLUG-SIDE and PUSH-WALL is indeed very high at the beginning, it rapidly deteriorates. This is an example of catastrophic forgetting. Since there is no data representing the 3rd and 4th tasks at the beginning of the training, the model instead uses its full capacity to learn the 1st and the 2nd task, causing interference.

### 4.3 CL METHODS

Next, we check whether continual learning methods that aim to mitigate forgetting, are able to improve performance in this setting. We repeat the same experiment, but this time we apply the L2, EWC, and behavioral cloning strategies described in Subsection 2.2 during fine-tuning. The results presented in Figure 5a show that EWC and behavioral cloning significantly outperform the fine-tuning baseline, both in terms of the final performance as well the speed of learning. L2 performs much worse which is not surprising as it is a very basic continual learning approach (Kirkpatrick et al., 2017).

From the perspective of investigating forgetting, the performance on the pre-trained tasks is of special importance. As shown in Figure 5b, continual learning methods are able to limit the impact of forgetting at the start of fine-tuning. Although the performance on these tasks initially dips, EWC and behavioral cloning are able to quickly relearn a high-performance policy, which suggests that not all of the knowledge was wiped out. In comparison, even after completing the whole training process, the success rate of vanilla fine-tuning stays low.

These results suggest that continual learning strategies might be useful tools for combating forgetting in fine-tuning reinforcement learning models. At the same time, we believe that even better solutions are possible. Continual learning methods tested here are fairly straightforward and were not built with this specific type of scenario in mind. We postulate that building continual learning-based algorithms with the specific aim of solving the fine-tuning problem might alleviate the issues related to the underlying data shift.

## 5 RELATED WORK

**Transfer in RL**  The prevailing paradigm in reinforcement learning is training models from scratch which is known to be largely inefficient. Training an agent even on the relatively simple Atari benchmark typically requires at least a few million training steps and the real-world problems are known to be much more complex than that (Ahn et al., 2022; Akkaya et al., 2019). Due to these difficulties, a new trend emerged where one tries to reuse and transfer knowledge as much as possible (Agarwal

et al., 2022). However, the fine-tuning strategy massively popular in supervised learning (Bommasani et al., 2021) is relatively less common in reinforcement learning. Instead, the community favors methods such as kickstarting (Schmitt et al., 2018; Lee et al., 2022a), and reusing offline data (Lee et al., 2022b; Kostrikov et al., 2021), skills (Pertsch et al., 2021) or the feature representations (Schwarzer et al., 2021; Stooke et al., 2021). In this study, we identified catastrophic forgetting as one of the problems making direct parameter reuse infeasible.

Numerous efforts in transfer learning were guided by the hope of achieving reliable foundation models in RL, thus replicating their success in other domains. Adaptive Agent Team et al. (2023) introduced an adaptive agent capable of in-context learning in previously unseen settings. Brohan et al. (2022) proposed Robotics Transformer which is able to perform a wide variety of manipulation tasks and exhibits favorable scaling properties. Reed et al. (2022) showed that a single model can be taught to perform a wide variety of tasks, including acting in multiple reinforcement learning environments

**Continual learning** Catastrophic forgetting is one of the core issues investigated in the field of continual learning, which deals with learning from a changing stream of data. In particular, there is a growing body of work at the intersection of continual and reinforcement learning (Khetarpal et al., 2022). Recent benchmarks include the robotics-based Continual World (Wołczyk et al., 2021), Lifelong Hanabi based on the multi-agent Hanabi game (Nekoei et al., 2021), and the CORA platform (Powers et al., 2022) which implements various baselines, metrics and testing environments. Several works proposed methods for continual reinforcement learning based on replay and distillation (Rolnick et al., 2019; Traoré et al., 2019) or modularity (Mendez et al., 2022; Gaya et al., 2022). Most of this research covers model-free approaches, but there has been a surge of interest in the previously under-explored problem of model-based continual reinforcement learning (Huang et al., 2021; Kessler et al., 2022). Although very relevant to our study, these works usually investigate changes in the dynamics of non-stationary environments. In this paper, we switch the perspective and focus on the data shifts occurring during training due to changes in the exploration policy.

## 6 LIMITATIONS & CONCLUSIONS

This study forms a preliminary investigation into the problem of forgetting in fine-tuning RL models. We show that fine-tuning a pre-trained model on compositional RL problems might result in a rapid deterioration of the performance of the pre-trained model if the relevant data is not available at the beginning of the training. This phenomenon is known as catastrophic forgetting. We show how it can occur in simple toyish situations (policy gradient with linear approximators on 1D gridworld) as well as more realistic problems (SAC with MLPs on a compositional robotic environment). Finally, we showed that applying CL methods significantly limits forgetting and allows for efficient transfer.

At the same time, this study, due to its preliminary nature, has numerous limitations which we hope to address in future work. We only considered a fairly strict formulation of the forgetting scenario where we assumed that the pre-trained model works perfectly on tasks that appear later in the fine-tuning phase. In practice, one should also consider the case when even though there are differences between the pre-training and fine-tuning tasks, the transfer is still possible. For example, if the model was pre-trained on picking and placing various objects, but in the fine-tuning phase it encounters a new type of object, the prior knowledge should still be useful for learning how to handle it. Additionally, our definition of a task is rigid and does not allow for continuous transitions between tasks. Finally, our experimental study only considered a single realistic environment. At the same time, even given these limitations, we see forgetting as an important problem to be solved and hope that addressing these issues in the future might help with building better foundation models in RL.

## ACKNOWLEDGEMENTS

The work of Maciej Wołczyk was supported by the National Centre of Science (Poland) Grant No. 2021/43/B/ST6/01456. The work of Bartłomiej Cupiał was supported by Foundation for Polish Science (grant no POIR.04.04.00-00-14DE/18-00 carried out within the Team-Net program co-financed by the European Union under the European Regional Development Fund). The work of Piotr Miłoś

was supported by the Polish National Science Center grant UMO-2017/26/E/ST6/00622 and UMO-2019/35/O/ST6/03464. We gratefully acknowledge Poland's high-performance computing infrastructure PLGrid (HPC Centers: ACK Cyfronet AGH) for providing computer facilities and support within computational grant no. PLG/2022/015875

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

| Method | actor reg. coef. | critic reg. coef. | memory |
|:---:|:---:|:---:|:---:|
| L2 | 1000 | 0 | - |
| EWC | 1 | 100 | - |
| BC | 1 | 0 | 10000 |

Table 1: Hyperparameters of CL methods

## A   TECHNICAL DETAILS

### A.1   CONTINUAL WORLD

In this paper, we use tasks from Continual World (Wołczyk et al., 2021). However, in order to adopt them to the fine-tuning scenario we introduce slight changes to the underlying environments. First of all, we used MetaWorld-v2 environments rather than MetaWorld-v1 used in Continual World, as they introduce more stable observation space (Yu et al., 2020). Additionally, we change the behavior of the terminal states. In the original paper, the environments are defined to run indefinitely, but during the training finite trajectories are sampled (i.e. 200 steps). On the 200-th step even though the trajectory ends, SAC receives information that the environment is still going. Effectively, it means that we still bootstrap our Q-value target as if this state was not terminal. This is a common approach for environments with infinite trajectories.

However, this approach is unintuitive from the perspective of task stitching. We would like to go from a given task to the next one at the moment when the success signal appears, without waiting for an arbitrary number of steps. As such, we introduce a change to the environments and terminate the episode in two cases: when the agent succeeds or when the time limit is reached. In both cases, SAC receives a signal that the state was terminal, which means no bootstrapping in the target Q-value. In order for the MDP to be fully observable, we append the normalized timestep to the state vector. Additionally, when the episode ends with success, we reward the agent with the "remaining" reward it would get until the end of the episode. That is, if the last reward was originally $r_t$, the augmented reward is given by $r'_t = \beta r_t (T - t)$. $\beta = 1.5$ is a coefficient to encourage the agent to succeed. Without the augmented reward there is a risk that the policy would avoid succeeding in order to get rewards for a longer period of time.

### A.2   SAC

We use the Soft Actor-Critic (Haarnoja et al., 2018a) algorithm for all the experiments on Continual World and use the same architecture as in the original paper (Wołczyk et al., 2021), which is a 4-layer MLP with 256 neurons each and Leaky-ReLU activations. We apply layer normalization after the first layer. The entropy coefficient is tuned automatically (Haarnoja et al., 2018b). We create a separate output head for each task in the neural networks and then we use the task ID information to choose the correct head. We found that this approach works better than adding the task ID to the observation vector.

For the base SAC, we started with the hyperparameters listed in Wołczyk et al. (2021), and then performed additional hyperparameter tuning. We set learning rate to $10^{-3}$ and use the Adam Kingma & Ba (2014) optimizer. The batch size is 128 in all experiments. We use L2, EWC, and BC as described in Wołczyk et al. (2021); Wolczyk et al. (2022). For each method, we perform a hyperparameter search on method-specific coefficients. That is, we look for the best-performing actor regularization weight and critic regularization weight. For BC we set the episodic memory buffer size to 10000. The final hyperparameters are listed in Table 1.

