# OpenReview forum: "On The Role of Forgetting in Fine-Tuning Reinforcement Learning Models"
_ICLR.cc/2023/Workshop/RRL — RRL 2023 Poster_

### Official Review · Reviewer_WHwK · 2023-02-27
**Good paper outlining catastrophic forgetting in fine-tuning pre-trained RL models**

**Rating:** 3
**Confidence:** 4

**Review:**

This paper aims to understand why foundation models in RL have not succeeded at the level that they have in fields outside RL. The authors claim that when fine-tuning pre-trained models on compositional RL problems, catastrophic forgetting occurs if the relevant parts of the state space are not available at the beginning of training. This claim first is supported by a toy experiment where an agent has to traverse a 1D gridworld to retrieve an apple and return to its original position, and where the model is pre-trained on only returning to the original position. Next, the authors create a less toyish environment using the tasks defined in Continual World that they call StitchedEnv. Here, new tasks start once previous ones are completed, and the model is pre-trained on the last two tasks. Finally, for the StitchedEnv environment, the authors propose incorporating methods from continual learning literature like L2 and Elastic Weight Consolidation, as well as performing behavioral cloning with buffers from pre-training. The results show the prevalence of catastrophic forgetting in fine-tuning pre-trained RL models, and the success of continual learning approaches in mitigating this problem.

The paper is well-written and clear. It communicates a potential wasteful problem in reusing prior computation and proposes clear methods to counter it. This is a good paper for the workshop.

---

### Official Review · Reviewer_E1Ni · 2023-02-27
**A study of fine-tuning issues with previously known conclusions**

**Rating:** 2
**Confidence:** 4

**Review:**

**Summary:**

The paper studies the problem of forgetting during fine-tuning of pre-trained models in RL.
The primary contributions are two-fold: first, authors analyze the phenomenon in a grid-world environment consisting of two stages, where the pre-training is done on the 2nd stage, and demonstrate that the harder the 1st stage is, the lower the knowledge retention about the optimal behavior on the 2nd stage is going to be.
The second contribution is an analysis of continual learning techniques for avoiding forgetting on a sequence of robotic manipulation tasks, where the pre-training is done on the last tasks of the sequence. The conclusion is that EWC and experience replay allow largely alleviating forgetting.
The authors argue that the problem setting studied might be relevant when dealing with foundation models in deep RL.


**Quality:**

The motivation of the paper is relevant to the scope of the workshop.
I enjoyed the minimalistic setting clearly isolating the problem.
The insights from the setting make sense: the more gradient updates on states outside of the pre-training dataset the agent makes, the lower the knowledge retention is on the states from the pre-training data.


**Clarity:**

Overall, the paper is clearly written and it is possible to understand all claims and details of experiments supporting the claims.
The reviewer recommends improving the rigor of statements and avoiding non-falsifiable phrases of the form: "high forgetting and overall poor performance."


**Significance:**

The significance of the paper is limited. The main issue is the research methodology: both of the settings are somewhat contrived and it is unclear to which extent the challenges in these settings would be encountered in practice.
The insight from the grid-world environment that the pre-training behavior gets forgotten is a known issue in the continual learning literature [1].
Likewise, the main practical conclusion is somewhat underwhelming: "continual learning techniques mostly address forgetting issues" does not give much insight.


Misc:
* I'd recommend avoiding the term "Compositional MDPs" --- it might be misleading as there's a vast literature on compositional generalization (e.g. [2])
* The huggingface link is not properly anonymized, suggesting that the submission is from https://parallax.co.in/

Overall recommendation:
The reviewer leans towards recommending the paper for acceptance to the workshop. It might spur a discussion at the workshop about the issues with fine-tuning of the pre-trained models. Extending results beyond the toy environment and a single sequence of control tasks would increase the significance of the paper.

[1] Kirkpatrick, James, Razvan Pascanu, Neil Rabinowitz, Joel Veness, Guillaume Desjardins, Andrei A. Rusu, Kieran Milan et al. "Overcoming catastrophic forgetting in neural networks." Proceedings of the national academy of sciences 114, no. 13 (2017): 3521-3526.

[2] Loula, Joao, Marco Baroni, and Brenden M. Lake. "Rearranging the familiar: Testing compositional generalization in recurrent networks." arXiv preprint arXiv:1807.07545 (2018).